# Health inequity associated with financial hardship among patients with kidney failure

**Marques Shek Nam Ng** **\*, Dorothy Ngo Sheung Chan, Winnie Kwok Wei So**

The Nethersole School of Nursing, Faculty of Medicine, The Chinese University of Hong Kong, Hong Kong, China

\* marquesng@cuhk.edu.hk

## Abstract

Financial hardship is a common challenge among patients with kidney failure and may have negative health consequences. Therefore, financial status is regarded as an important determinant of health, and its impact needs to be investigated. This cross-sectional study aimed to identify the differences in patient-reported and clinical outcomes among kidney failure patients with different financial status. A total of 354 patients with kidney failure were recruited from March to June 2017 at two hospitals in Hong Kong. The Dialysis Symptoms Index and Kidney Disease Quality of Life-36 were used to evaluate patient-reported outcomes. Clinical outcomes were retrieved from medical records and assessed using the Karnofsky Performance Scale (functional status) and Charlson Comorbidity Index (comorbidity level). Patients were stratified using two dichotomised variables, employment status and income level, and their outcomes were compared using independent sample t-tests and Mann-Whitney U-tests. In this sample, the employment rate was 17.8% and the poverty rate was 61.2%. Compared with other patients, increased distress of specific symptoms and higher healthcare utilization, in terms of more emergency room visits and longer hospital stays, were found in patients with poorer financial status. Low-income patients reported a decreased mental quality of life. Financially underprivileged patients experienced health inequity in terms of impaired outcomes. Attention needs to be paid to these patients by providing financial assessments and interventions. Additional research is warranted to confirm these findings and understand the experience of financial hardship and health equity.

## Introduction

The global burden of kidney failure is increasing. More than 2 million patients have been diagnosed and treated with life-sustaining dialysis therapy worldwide [1]. In some countries, 2–3% of healthcare expenditures are used for managing kidney failure, and the demand for dialysis continues to surge [2]. At the individual level, patients often experience financial hardship associated with high treatment costs, unemployment, and reduced income. Although many countries provide reimbursement for dialysis therapy, patients are required to cover 12–71% of the costs out of pocket [3]. In addition, these patients have reduced productivity associated with disease-related disabilities. According to an international survey [4], less than 55% of

**Data Availability Statement:** Data cannot be shared publicly because of privacy issues. Confidential data are available from the CUHK Research Data Repository for researchers whose

work has been approved by an institutional review board. Request may be sent together with the research proposal and ethical approval to the corresponding author or the Repository (website: https://researchdata.cuhk.edu.hk / email: data@cuhk.edu.hk).

**Funding:** The authors received no specific funding for this work.

**Competing interests:** The authors have declared that no competing interests exist.

haemodialysis (HD) and 68% of peritoneal dialysis (PD) patients are employed. Limited income due to decreased employment may amplify financial hardship among these patients [5].

Financial hardship is a profound and significant determinant of health. In fact, higher burdens associated with kidney failure are found in countries that are less socioeconomically developed [1]. From an individual perspective, patients utilize their personal resources to pay for their medical and other daily expenses, which may lead to the depletion of financial reserves or incurrence of debt [6]. Evidence suggests that impaired financial well-being is associated with poor physical and psychological health in patients with chronic illnesses [7, 8]. Negative health outcomes, including depression, anxiety, lower health-related quality of life (HRQoL), and higher mortality risk, have been reported in patients with kidney failure who are socioeconomically disadvantaged [9–11]. These outcomes may be seen as the consequences of health inequity caused by social determinants, especially financial factors.

Despite the impact of financial hardship on health, few studies have evaluated the relationships between this hardship and various health outcomes among patients with kidney failure [12]. A better understanding of these relationships may advance our understanding of health inequity among patients with kidney failure. Hence, in addition to financial aid, kidney care providers can proactively identify financially underprivileged patients and provide specific interventions that promote equal participation in daily life [13]. In Hong Kong, health consequences of financial hardship in patients with kidney failure have been rarely examined. Therefore, to explore directions for future research, the aim of this study was to identify the differences in patient-reported and clinical outcomes between kidney failure patients with high and low financial status.

This study was conducted in Hong Kong, one of the financial centres in the Chinese territory. While this city has a relatively high per capita gross domestic product of 49,801 USD in 2021 [14], it is famous for the large wealth gap. Its latest Gini coefficient in 2019 reached 51.8, which reflected fair inequality within this 700-million population [15]. In terms of the healthcare system, Hong Kong has a predominant public sector that provides over 90% of inpatient services and is largely subsidised by taxation [16]. Despite the availability of subsidised services, patients with kidney failure are required to use out-of-pocket expenses to cover costs of dialysis consumables and self-financed drugs.

## Materials and methods

The cross-sectional data of 354 patients from a mixed-methods study were analysed [17]. This sample size was estimated to generate sufficient data for the latent class analysis in the original study. These patients were recruited at the dialysis clinics of Pamela Youde Nethersole Eastern Hospital and United Christian Hospital from March to June 2017. These hospitals were serving populations with highest and lowest household incomes in Hong Kong [18]. The inclusion criteria included: 1) adults diagnosed with kidney failure; 2) received any modality of dialysis therapy for three or more consecutive months; and 3) were willing to provide written consent. Those with active psychiatric disorders (e.g., schizophrenia, dementia) were excluded. Given that the complete patient lists could not be generated due to privacy issues, a convenience sampling method was used.

A research assistant approached patients in the clinics and screened for eligibility. After explaining the study and obtaining informed consent, the research assistant administered a questionnaire containing a demographic form (see S1 File) and the instruments. Then, electronic health records were reviewed at the dialysis clinics. This study was approved by the Joint CUHK-NTEC Clinical Research Ethics Committee (reference number: 2017.092),

HKEC Research Ethics Committee (reference number: HKEC-2017-008), and KCC/KEC Research Ethics Committee (reference number: KC/KE-17-0016/ER-3) prior to data collection. Written consent was obtained from all participants.

## Instruments

Symptoms and HRQoL were selected as the patient-reported outcomes. The Dialysis Symptoms Index (DSI) was used to assess the distress levels of 30 symptoms experienced in the past month [19]. A higher score indicates a higher level of distress. The Chinese version of the DSI demonstrated excellent content validity (0.99) and internal consistency (α = 0.87) [20]. The Kidney Disease Quality of Life-36 was used to evaluate the HRQoL of patients [21]. It consists of 24 disease-specific and 12 generic questions that can be divided into three subscales (symptom, burden, and effect of kidney disease) and two summary scores (physical and mental component summary). A higher score indicates better performance in the specific domain. Its Chinese version demonstrated good test-retest reliability (interclass correlation coefficient = 0.79–0.92) and acceptable internal consistency (α = 0.60–0.93) [22].

The clinical outcomes included the functional status, comorbidity level, healthcare service utilization, and biochemical parameters. The Karnofsky Performance Scale was used to evaluate functional status [23]. A higher score indicates higher ability to perform activities of daily living (range: 0–100). The Charlson Comorbidity Index was used to assess the comorbidity level [24]. Patients' relative burden of comorbidity is evaluated based on the weighted sum of 14 conditions (range: 0–33). Other clinical data, including healthcare service utilization in past six months (such as number of emergency room [ER] attendance and days of hospital stay) and biochemical parameters (such as serum albumin concentration), were retrieved from the patients' electronic health records. Glomerular filtration rate was estimated based on the Modification of Diet in Renal Disease (MDRD) equation [25]. These outcomes were analysed as continuous variables.

## Analyses

Background characteristics and outcomes were summarised using appropriate descriptive statistics (e.g., percentage, mean, median) depending on the distributions of variables. Two dichotomized variables that reflected the patient's financial status were created: employment status (unemployed vs. employed) and income level (below vs. above poverty line). In terms of income level, those earning a monthly household income of ≤ 10,000 HKD (approximately 1,290 USD) were regarded as below the poverty line, which is in line with the government's definition [18]. After stratifying the patients by these variables, background characteristics were compared using chi-squared tests (sex, marital status, education level, dialysis modality, history of transplantation), independent sample t-tests and Mann-Whitney U-tests, as appropriate.

Unadjusted comparisons on patient-reported and clinical outcomes between groups of the financial status variables were made using independent sample t-tests and Mann-Whitney U-tests for normally and skewedly distributed variables, respectively. Then, multiple regression analyses were conducted to compare the outcomes between groups with adjustment for participants' background characteristics (sex, marital status, education level, dialysis modality, history of transplantation, age, duration on dialysis). All analyses were conducted using SPSS version 25.0 (IBM Corp., Armonk, NY). A two-sided p-value of < 0.05 was considered as statistically significant.

## Results

Of the 424 patients approached, 22 did not meet eligibility and 48 declined to participate. Among the consented patients, 58.5% were male. The patients had a mean age of 60.9 years

(Table 1). Most of the patients received PD (69.9%) and had been on dialysis for a median of 36 months (inter-quartile range: 17–60). Overall, 17.8% of the patients were employed, and 61.2% were below the poverty line. The education level significantly differentiated patients regardless employment status and income level (p ≤ 0.001). Compared with other patients, a larger proportion of employed patients had received kidney transplants (9.5%; p = 0.035), and a larger proportion of patients above the poverty line were married (76.6%; p = 0.01).

Tables 1 and 2 present the differences in patient-reported and clinical outcomes between groups. Compared with employed patients, in unadjusted analyses, those who were unemployed reported higher levels of tiredness (mean: 2.33), joint or bone pain (mean: 1.60), and trouble falling asleep (mean: 2.13) (all p ≤ 0.05). Adjusted analyses showed that employed patients were associated with lower level of tiredness (Regression coefficient B = -0.543; standard error [SE] = 0.250; p = 0.031), and joint/bone pain (B = -0.527; SE = 0.267; p = 0.049) than those unemployed patients after adjusting for background characteristics. While no significant relationship was found between employment status and quality of life outcomes in the adjusted analyses, those employed reported a significantly lower mean number of ER attendance in the past six months (B = -0.384; SE = 0.171; p = 0.025) than their unemployed counterparts.

Patients below the poverty line reported higher levels of dry mouth (mean: 1.63), dry skin (mean: 2.68), itching (mean: 2.76), and trouble staying asleep (mean: 2.07) than those above the poverty line (all p ≤ 0.05). Interestingly, patients with lower income had less severe sexual symptoms (all p ≤ 0.05), namely a decreased interest in sex (mean: 0.41) and difficulty becoming sexually aroused (mean: 0.37). However, in the adjusted analyses, only dry mouth (B = -0.385; SE = 0.186; p = 0.039), dry skin (B = -0.826; SE = 0.204; p < 0.001), and itching (B = -0.536; SE = 0.209; p = 0.011) remain significantly different between the two income level groups. In addition, patients above poverty line had lower KDQOL-36 Mental Component Summary scale scores (B = 2.725; SE = 1.280; p = 0.034) and shorter hospital stays in the past six months (B = -2.810; SE = 1.367; p = 0.041) when compared with those who were below poverty line in the adjusted analyses.

In terms of clinical outcomes, patients with a poorer financial status had higher comorbidity levels (all p ≤ 0.05). Patients below the poverty line had a lower serum albumin concentration (p = 0.004) than patients with higher earnings. However, these results were not statistically significant after adjusting for background characteristics.

## Discussion

The findings from this study suggest that patient-reported and clinical outcomes differ between patients with different financial statuses in terms of their employment and income level. Based on our preliminary findings, while no significant relationship was found between financial status and most patient-reported outcomes, patients who were unemployed or living below the poverty line reported higher distress associated with specific symptoms and more health care utilization than other patients. Consistent with existing evidence [8–10], the impact of financial hardship on health disparities among patients with kidney failure warrants additional attention.

Compared with the general population in Hong Kong, the employment rate in this study was halved and the poverty rate was three times higher (cf. employment rate: 34.9%; poverty rate: 21.4%) [18]. This finding is an alarming sign that in this city, which is well known for economic inequality, financial hardship is very common among patients with kidney failure. Our findings indicate the negative impact of such hardship within this group. Consistent with our previous studies [8, 26], financially underprivileged patients may experience a higher symptom

**Table 1. Background characteristics and comparison between different employment groups.**

| | Overall (N = 354) | | Unemployed (n = 291; 82.2%) | | Employed (n = 63; 17.8%) | | Unadjusted analyses | Adjusted analyses^ | | |
|---|---|---|---|---|---|---|---|---|---|---|
| | | | | | | | p | B | SE | p |
| *Background characteristics* | | | | | | | | | | |
| Male (vs. Female)^Ψ | 207 | 58.5% | 164 | 56.4% | 43 | 68.3% | 0.082 | | | |
| Married (vs. Not married)^Ψ | 243 | 68.6% | 206 | 70.8% | 37 | 58.7% | 0.061 | | | |
| Secondary education (vs. Primary education or below)^Ψ | 231 | 65.3% | 177 | 60.8% | 54 | 85.7% | <0.001* | | | |
| Peritoneal dialysis (vs. Haemodialysis)^Ψ | 255 | 72.0% | 212 | 72.9% | 43 | 68.3% | 0.461 | | | |
| History of transplantation (vs. No history of transplantation)^Ψ | 16 | 4.5% | 10 | 3.4% | 6 | 9.5% | 0.035* | | | |
| Age (years) | 60.93 | 11.89 | 62.70 | 11.33 | 52.63 | 11.00 | <0.001* | | | |
| Month on dialysis^# | 36 | 17–60 | 36 | 18–60 | 30 | 11–60 | 0.309 | | | |
| *Patient-reported outcomes* | | | | | | | | | | |
| Dialysis Symptoms Index | 34.16 | 23.03 | 36.07 | 22.03 | 31.32 | 24.73 | 0.163 | -4.617 | 3.363 | 0.171 |
| Constipation | 1.09 | 1.60 | 1.15 | 1.65 | 0.78 | 1.35 | 0.057 | -0.196 | 0.244 | 0.422 |
| Chest pain | 0.50 | 1.16 | 0.54 | 1.18 | 0.33 | 1.06 | 0.203 | -0.301 | 0.177 | 0.090 |
| Nausea | 0.79 | 1.39 | 0.74 | 1.35 | 1.02 | 1.54 | 0.190 | 0.095 | 0.209 | 0.649 |
| Vomiting | 0.65 | 1.34 | 0.64 | 1.33 | 0.70 | 1.40 | 0.765 | -0.109 | 0.205 | 0.596 |
| Diarrhoea | 0.71 | 1.34 | 0.70 | 1.36 | 0.73 | 1.30 | 0.876 | -0.057 | 0.200 | 0.776 |
| Decreased appetite | 1.18 | 1.55 | 1.21 | 1.57 | 1.05 | 1.49 | 0.453 | -0.040 | 0.236 | 0.867 |
| Cramps | 1.43 | 1.62 | 1.41 | 1.62 | 1.54 | 1.62 | 0.552 | 0.198 | 0.243 | 0.417 |
| Oedema | 0.97 | 1.36 | 0.93 | 1.35 | 1.14 | 1.40 | 0.271 | 0.215 | 0.207 | 0.299 |
| Shortness of breath | 1.07 | 1.49 | 1.09 | 1.51 | 1.00 | 1.45 | 0.668 | -0.181 | 0.228 | 0.428 |
| Dizziness | 0.96 | 1.46 | 0.99 | 1.46 | 0.83 | 1.50 | 0.430 | -0.120 | 0.224 | 0.593 |
| Restless legs | 0.60 | 1.32 | 0.64 | 1.36 | 0.43 | 1.10 | 0.243 | -0.335 | 0.201 | 0.096 |
| Limb numbness | 1.04 | 1.53 | 1.08 | 1.55 | 0.87 | 1.44 | 0.342 | -0.103 | 0.233 | 0.659 |
| Tiredness | 2.23 | 1.66 | 2.33 | 1.65 | 1.79 | 1.65 | 0.020* | -0.543 | 0.250 | 0.031* |
| Coughing | 1.35 | 1.56 | 1.40 | 1.60 | 1.14 | 1.34 | 0.195 | -0.223 | 0.239 | 0.651 |
| Dry mouth | 1.49 | 1.56 | 1.54 | 1.58 | 1.24 | 1.43 | 0.164 | -0.158 | 0.235 | 0.501 |
| Joint/bone pain | 1.51 | 1.75 | 1.60 | 1.75 | 1.08 | 1.69 | 0.033* | -0.527 | 0.267 | 0.049* |
| Headache | 0.73 | 1.29 | 0.76 | 1.30 | 0.60 | 1.28 | 0.352 | -0.273 | 0.195 | 0.162 |
| Muscle soreness | 1.25 | 1.55 | 1.30 | 1.56 | 1.03 | 1.48 | 0.220 | -0.251 | 0.235 | 0.286 |
| Difficulty concentrating | 1.05 | 1.50 | 1.08 | 1.54 | 0.89 | 1.35 | 0.355 | -0.234 | 0.230 | 0.309 |
| Dry skin | 2.42 | 1.73 | 2.42 | 1.78 | 2.40 | 1.51 | 0.918 | 0.293 | 0.258 | 0.256 |
| Itching | 2.58 | 1.74 | 2.66 | 1.76 | 2.21 | 1.62 | 0.063 | -0.212 | 0.264 | 0.422 |
| Worrying | 1.05 | 1.56 | 1.06 | 1.57 | 1.02 | 1.55 | 0.845 | -0.015 | 0.233 | 0.950 |
| Feeling nervous | 0.88 | 1.46 | 0.90 | 1.48 | 0.79 | 1.45 | 0.592 | -0.118 | 0.218 | 0.591 |
| Trouble falling asleep | 2.02 | 1.98 | 2.13 | 1.99 | 1.51 | 1.87 | 0.024* | -0.455 | 0.299 | 0.129 |
| Trouble staying asleep | 1.92 | 1.88 | 2.00 | 1.89 | 1.54 | 1.76 | 0.080 | -0.286 | 0.284 | 0.316 |
| Feeling agitated | 1.03 | 1.52 | 1.06 | 1.54 | 0.89 | 1.50 | 0.426 | -0.163 | 0.228 | 0.475 |
| Feeling sad | 0.86 | 1.43 | 0.92 | 1.47 | 0.60 | 1.21 | 0.076 | -0.232 | 0.213 | 0.278 |
| Feeling anxious | 0.78 | 1.35 | 0.79 | 1.35 | 0.75 | 1.38 | 0.828 | -0.041 | 0.206 | 0.842 |
| Decreased interest in sex | 0.56 | 1.28 | 0.54 | 1.29 | 0.70 | 1.20 | 0.361 | -0.178 | 0.186 | 0.339 |
| Difficulty getting sexually aroused | 0.53 | 1.27 | 0.48 | 1.25 | 0.73 | 1.33 | 0.164 | -0.071 | 0.185 | 0.703 |
| Kidney Disease Quality of Life 36 | | | | | | | | | | |
| Symptom of kidney disease | 78.20 | 15.02 | 77.90 | 14.68 | 46.60 | 16.55 | 0.417 | 1.308 | 2.263 | 0.564 |
| Effect of kidney disease | 71.73 | 19.45 | 72.52 | 19.53 | 68.11 | 18.84 | 0.103 | -2.280 | 2.822 | 0.420 |
| Burden of kidney disease | 30.54 | 21.70 | 30.26 | 22.24 | 31.85 | 19.14 | 0.600 | -0.441 | 3.300 | 0.894 |
| Physical Component Summary | 37.87 | 9.54 | 37.27 | 9.61 | 40.65 | 8.74 | 0.011* | 2.285 | 1.437 | 0.113 |

*(Continued)*

**Table 1.** (Continued)

| | Overall | | Unemployed | | Employed | | Unadjusted analyses | Adjusted analyses^ | | |
|---|---|---|---|---|---|---|---|---|---|---|
| | (N = 354) | | (n = 291; 82.2%) | | (n = 63; 17.8%) | | | | | |
| Mental Component Summary | 48.54 | 10.84 | 48.80 | 11.05 | 47.36 | 9.86 | 0.340 | -1.500 | 1.618 | 0.355 |
| *Clinical Outcomes* | | | | | | | | | | |
| Karnofsky Performance Scale | 88.59 | 13.43 | 87.56 | 14.07 | 93.33 | 8.61 | <0.001* | 1.687 | 1.940 | 0.385 |
| Charlson Comorbidity Index# | 2. | 0–3 | 2 | 0–3 | 0 | 0–2 | <0.001* | -0.488 | 0.299 | 0.103 |
| Clinical visits in 6 months# | 0 | 0–2 | 0 | 0–2 | 0 | 0–2 | 0.241 | 0.602 | 0.339 | 0.077 |
| ER attendance in 6 months# | 0 | 0–1 | 0 | 0–1 | 0 | 0–0 | 0.002* | -0.384 | 0.171 | 0.025* |
| Days of hospital stay in 6 months# | 0 | 0–7 | 0 | 0–7 | 0 | 0–6 | 0.942 | 1.490 | 1.724 | 0.388 |
| Glomerular filtration rate (mL/min/1.73m$^2$) | 5.45 | 2.26 | 5.50 | 2.32 | 5.21 | 1.95 | 0.359 | -0.021 | 0.336 | 0.951 |
| Serum albumin (g/L) | 35.96 | 4.68 | 35.73 | 4.74 | 36.98 | 4.28 | 0.054 | 0.258 | 0.654 | 0.694 |
| Serum calcium (mmol/L) | 2.28 | 0.19 | 2.28 | 0.20 | 2.28 | 0.16 | 0.973 | -0.020 | 0.029 | 0.490 |
| Serum phosphate (mmol/L) | 1.78 | 0.55 | 1.77 | 0.56 | 1.82 | 0.54 | 0.548 | -0.132 | 0.081 | 0.103 |
| Haemoglobin (g/dL) | 10.27 | 2.79 | 10.31 | 2.98 | 10.12 | 1.67 | 0.631 | -0.298 | 0.434 | 0.492 |

*p < 0.05

Data of variables marked with

Ψ are presented as frequency and percentage

# as median and inter-quartile range, and all others as mean and standard deviation.

^In the adjusted analyses, demographic factors adjusted included: Sex, marital status, education level, dialysis modality, history of transplantation, age, and duration on dialysis

Abbreviations: B = Regression coefficient of the dummy variable (Employment: 0 = Unemployed; 1 = Employed); SD = Standard deviation; SE = Standard error of the regression coefficient

burden in terms of tiredness, dry mouth, skin problems, and pain. There are some possible reasons for this relationship. As reported in one study [27], financially underprivileged patients may have fewer resources to pay for healthcare services, especially preventive care. They directly turn to the highly subsidized hospital care when they cannot tolerate their symptoms [28]. This may explain why patients with impaired financial well-being reported poorer outcomes, particularly a higher mean number of ER visits and length of hospital stay. Therefore, kidney care providers need to pay special attention to the financial needs of patients by incorporating appropriate assessments and interventions in routine care [12]. Nevertheless, there is a need to examine equity in the healthcare system to ensure that essential services are provided regardless of a person's financial status. In addition, there may be a reciprocal relationship between financial hardship and outcomes. Symptoms and other outcomes may be signs of deteriorating health associated with financial hardship. However, patients with a poorer symptom status or poorer outcomes may also have a higher demand for healthcare services and a lower physical capacity for engaging actively in employment [29]. These consequences lead to increased medical expenditures and decreased income, which eventually intensify financial hardship [6]. A better understanding of the experience of financial hardship is warranted to explore the factors that modulate the relationship between financial hardship and health.

It is noteworthy that despite no significant relationship being found between financial status, quality of life outcomes, and biochemical parameters, the impact of financial hardship on health should not be underestimated. Associations between financial hardship and various symptoms, such as depression, fatigue, and pain, have been found in previous studies [8]. One

**Table 2. Comparison between different income groups.**

| | Below Poverty Line | | Above Poverty Line | | Unadjusted analyses | Adjusted analyses^ | | |
|---|---|---|---|---|---|---|---|---|
| | (n = 217; 61.3%) | | (n = 137; 38.7%) | | | | | |
| *Background characteristics* | | | | | *p* | *B* | *SE* | *p* |
| Male (vs. Female)<sup>Ψ</sup> | 128 | 59.0% | 79 | 36.4% | 0.806 | | | |
| Married (vs. Not married)<sup>Ψ</sup> | 138 | 63.6% | 105 | 76.6% | 0.010* | | | |
| Secondary education (vs. Primary education or below)<sup>Ψ</sup> | 127 | 58.5% | 104 | 75.9% | 0.001* | | | |
| Peritoneal dialysis (vs. Haemodialysis)<sup>Ψ</sup> | 158 | 72.8% | 97 | 70.8% | 0.682 | | | |
| History of transplantation (vs. No history of transplantation)<sup>Ψ</sup> | 10 | 4.6% | 6 | 4.4% | 0.920 | | | |
| Age (years) | 63.36 | 11.51 | 57.10 | 11.51 | <0.001* | | | |
| Month on dialysis<sup>#</sup> | 36 | 20–60 | 30 | 12–55 | 0.081 | | | |
| *Patient-reported outcomes* | | | | | | | | |
| Dialysis Symptoms Index | 36.18 | 21.75 | 33.69 | 23.81 | 0.323 | -3.615 | 2.660 | 0.175 |
| Constipation | 1.21 | 1.67 | 0.89 | 1.48 | 0.059 | -0.164 | 0.193 | 0.396 |
| Chest pain | 0.53 | 1.21 | 0.46 | 1.09 | 0.070 | -0.070 | 0.140 | 0.618 |
| Nausea | 0.72 | 1.34 | 0.89 | 1.47 | 0.282 | -0.102 | 0.165 | 0.537 |
| Vomiting | 0.61 | 1.27 | 0.72 | 1.45 | 0.485 | -0.086 | 0.163 | 0.599 |
| Diarrhoea | 0.76 | 1.43 | 0.61 | 1.20 | 0.283 | -0.279 | 0.158 | 0.078 |
| Decreased appetite | 1.21 | 1.52 | 1.14 | 1.60 | 0.685 | -0.067 | 0.187 | 0.722 |
| Cramps | 1.43 | 1.62 | 1.43 | 1.62 | 0.991 | -0.035 | 0.193 | 0.855 |
| Oedema | 0.98 | 1.38 | 0.96 | 1.33 | 0.864 | -0.174 | 0.164 | 0.289 |
| Shortness of breath | 1.06 | 1.51 | 1.09 | 1.48 | 0.888 | 0.016 | 0.180 | 0.931 |
| Dizziness | 0.97 | 1.45 | 0.94 | 1.49 | 0.870 | -0.075 | 0.177 | 0.670 |
| Restless legs | 0.54 | 1.21 | 0.71 | 1.47 | 0.261 | 0.111 | 0.159 | 0.486 |
| Limb numbness | 1.06 | 1.53 | 1.00 | 1.54 | 0.700 | 0.140 | 0.184 | 0.446 |
| Tiredness | 2.21 | 1.64 | 2.28 | 1.70 | 0.700 | 0.136 | 0.198 | 0.492 |
| Coughing | 1.39 | 1.61 | 1.29 | 1.47 | 0.576 | -0.086 | 0.189 | 0.651 |
| Dry mouth | 1.63 | 1.60 | 1.26 | 1.46 | 0.027* | -0.385 | 0.186 | 0.039* |
| Joint/bone pain | 1.55 | 1.74 | 1.43 | 1.77 | 0.522 | 0.006 | 0.211 | 0.978 |
| Headache | 0.76 | 1.35 | 0.68 | 1.20 | 0.543 | -0.108 | 0.154 | 0.483 |
| Muscle soreness | 1.27 | 1.53 | 1.22 | 1.57 | 0.775 | -0.062 | 0.186 | 0.738 |
| Difficulty concentrating | 1.05 | 1.50 | 1.04 | 1.51 | 0.967 | -0.040 | 0.182 | 0.828 |
| Dry skin | 2.68 | 1.68 | 1.99 | 1.73 | <0.001* | -0.826 | 0.204 | <0.001* |
| Itching | 2.76 | 1.75 | 2.28 | 1.70 | 0.010* | -0.536 | 0.209 | 0.011* |
| Worrying | 1.05 | 1.59 | 1.05 | 1.52 | 0.998 | -0.108 | 0.184 | 0.558 |
| Feeling nervous | 0.89 | 1.51 | 0.88 | 1.42 | 0.933 | -0.064 | 0.173 | 0.709 |
| Trouble falling asleep | 2.12 | 1.99 | 1.85 | 1.97 | 0.200 | -0.149 | 0.237 | 0.530 |
| Trouble staying asleep | 2.07 | 1.86 | 1.66 | 1.88 | 0.045* | -0.336 | 0.225 | 0.136 |
| Feeling agitated | 1.11 | 1.55 | 0.91 | 1.50 | 0.229 | -0.246 | 0.181 | 0.175 |
| Feeling sad | 0.95 | 1.48 | 0.72 | 1.34 | 0.138 | -0.258 | 0.168 | 0.127 |
| Feeling anxious | 0.81 | 1.36 | 0.74 | 1.35 | 0.640 | -0.203 | 0.163 | 0.215 |
| Decreased interest in sex | 0.41 | 1.08 | 0.82 | 1.51 | 0.003* | 0.215 | 0.147 | 0.144 |
| Difficulty getting sexually aroused | 0.37 | 1.51 | 0.77 | 1.52 | 0.004* | 0.219 | 0.147 | 0.136 |
| Kidney Disease Quality of Life 36 | | | | | | | | |
| Symptom of kidney disease | 77.84 | 15.07 | 78.77 | 14.99 | 0.571 | 1.556 | 1.790 | 0.385 |
| Effect of kidney disease | 71.79 | 19.13 | 71.65 | 20.03 | 0.947 | 4.210 | 2.232 | 0.060 |
| Burden of kidney disease | 29.46 | 22.07 | 32.25 | 21.06 | 0.239 | 2.379 | 2.611 | 0.363 |
| Physical Component Summary | 37.51 | 9.56 | 38.45 | 9.50 | 0.367 | 0.252 | 1.137 | 0.825 |

*(Continued)*

**Table 2.** (Continued)

| | Below Poverty Line | | Above Poverty Line | | Unadjusted analyses | Adjusted analyses^ | | |
|---|---|---|---|---|---|---|---|---|
| | (n = 217; 61.3%) | | (n = 137; 38.7%) | | | | | |
| Mental Component Summary | 47.88 | 10.98 | 49.58 | 10.59 | 0.151 | 2.725 | 1.280 | 0.034* |
| *Clinical Outcomes* | | | | | | | | |
| Karnofsky Performance Scale | 87.65 | 13.79 | 90.07 | 12.75 | 0.098 | 0.024 | 1.534 | 0.987 |
| Charlson Comorbidity Index# | 2 | 0–3 | 1 | 0–2 | 0.001* | -0.251 | 0.236 | 0.289 |
| Clinical visits in 6 months# | 0 | 0–1 | 0 | 0–2 | 0.136 | -0.017 | 0.268 | 0.949 |
| ER attendance in 6 months# | 0 | 0–1 | 0 | 0–1 | 0.044* | -0.233 | 0.136 | 0.088 |
| Days of hospital stay in 6 months# | 0 | 0–11 | 0 | 0–5 | 0.062 | -2.810 | 1.367 | 0.041* |
| Glomerular filtration rate (mL/min/1.73m$^2$) | 5.53 | 2.25 | 5.32 | 2.27 | 0.397 | -0.088 | 0.267 | 0.741 |
| Serum albumin (g/L) | 35.39 | 4.83 | 36.85 | 4.30 | 0.004* | 0.969 | 0.519 | 0.063 |
| Serum calcium (mmol/L) | 2.27 | 0.20 | 2.29 | 0.19 | 0.302 | 0.019 | 0.023 | 0.395 |
| Serum phosphate (mmol/L) | 1.74 | 0.55 | 1.84 | 0.56 | 0.113 | 0.048 | 0.064 | 0.453 |
| Haemoglobin (g/dL) | 10.29 | 1.68 | 10.24 | 3.94 | 0.878 | -0.028 | 0.344 | 0.935 |

*p < 0.05

Data of variables marked with

Ψ are presented as frequency and percentage

# as median and inter-quartile range, and all others as mean and standard deviation.

^In the adjusted analyses, demographic factors adjusted included: Sex, marital status, education level, dialysis modality, history of transplantation, age, and duration on dialysis

Abbreviations: B = Regression coefficient of the dummy variable (Income: 0 = Below poverty line; 1 = Above poverty line); SD = Standard deviation; SE = Standard error of the regression coefficient

reason for the lack of statistical significance is the limitations in evaluating financial status. As concluded in a systematic review [7], financial hardship consists of material, psychological, and behavioural changes in relation to financial status. In this study, financial hardship is indicated by unemployment and low income, which only reflect some aspects of the material change. In fact, some patients who are unemployed or with low income may be abundantly supported by families, personal savings, or other financial aids. Hence, the selected indicators may not be specific and sensitive enough to evaluate financial hardship and its impact.

Our preliminary evidence reveals the consequences of health inequity and suggests directions for researching issues associated with financial hardship among patients with kidney failure. However, several limitations warrant consideration. Although cross-sectional data were analysed and limited variables were evaluated, other factors that might confound the relationship between financial hardship and outcomes (e.g., financial aids, home ownership, personal savings, health spendings) were not assessed and controlled [8]. Given these methodological limitations, findings should be interpreted as markers of potential influences of financial status on patient outcomes. In addition, the causal relationships among financial statuses, kidney failure, incapacity for employment, and health outcomes need to be examined using a longitudinal design. Of note, as mentioned above, financial status was evaluated in terms of employment status and income level only, which might inadequately reflect the full picture of financial well-being. Therefore, the following suggestions are made: 1) factors influencing financial hardship should be identified and controlled in further analyses; 2) a longitudinal study should be conducted to evaluate changes in financial status and outcomes; and 3) a comprehensive conceptualization of financial hardship should be adopted.

## Conclusion

Financial hardship is very common among patients with kidney failure, especially in Hong Kong, in terms of high percentages of unemployment and poverty. Our preliminary evidence suggests that this hardship may result in health inequity and manifest in impaired patient-reported and clinical outcomes. However, given the methodological limitations, additional research is warranted to confirm these findings and understand the experience of financial hardship and health equity.

## Supporting information

**S1 File. Demographic form.** This form was developed by the research team to collect demographic information of the participants.
(PDF)

## Acknowledgments

The authors would like to express their gratitude to Dr Stephen Mo, Ms Eva Ho, Dr Sunny Wong, Ms Yun Ho Hui, and the staff members of the study sites for their assistance in data collection. Special thanks are given to Dr Kai Chow Choi for his professional statistical support.

## Author Contributions

**Conceptualization:** Marques Shek Nam Ng, Winnie Kwok Wei So.

**Data curation:** Marques Shek Nam Ng.

**Formal analysis:** Marques Shek Nam Ng, Dorothy Ngo Sheung Chan, Winnie Kwok Wei So.

**Investigation:** Marques Shek Nam Ng.

**Methodology:** Marques Shek Nam Ng, Winnie Kwok Wei So.

**Project administration:** Marques Shek Nam Ng.

**Writing – original draft:** Marques Shek Nam Ng.

**Writing – review & editing:** Dorothy Ngo Sheung Chan, Winnie Kwok Wei So.

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
