## [Decision Letter · Decision Letter 0]

22 Nov 2022

PONE-D-22-21227Health Inequity Associated with Financial Hardship Among Patients with

End-stage Kidney Disease: A Secondary AnalysisPLOS ONE

Dear Dr. Ng,

Thank you for submitting your manuscript to PLOS ONE. After careful consideration, we feel that it has merit but does not fully meet PLOS ONE’s publication criteria as it currently stands. Therefore, we invite you to submit a revised version of the manuscript that addresses the points raised during the review process.

Please note that we have only been able to secure a single reviewer to assess your manuscript. We are issuing a decision on your manuscript at this point to prevent further delays in the evaluation of your manuscript. Please be aware that the editor who handles your revised manuscript might find it necessary to invite additional reviewers to assess this work once the revised manuscript is submitted. However, we will aim to proceed on the basis of this single review if possible.

The reviewer has identified many aspects of your study design, methods and statistical analyses that require clarification and elaboration in order to demonstrate that your submission meets our third and fourth publication criteria (https://journals.plos.org/plosone/s/criteria-for-publication). Please respond carefully to all the points they have raised when preparing your revisions.

We look forward to receiving your revised manuscript.

Kind regards,

Jamie Males

Editorial Office

PLOS ONE

Journal Requirements:

Reviewers' comments:

Reviewer's Responses to Questions

**Comments to the Author**

1. Is the manuscript technically sound, and do the data support the conclusions?

Reviewer #1: No

2. Has the statistical analysis been performed appropriately and rigorously? 

Reviewer #1: No

3. Have the authors made all data underlying the findings in their manuscript fully available?

Reviewer #1: No

4. Is the manuscript presented in an intelligible fashion and written in standard English?

Reviewer #1: Yes

5. Review Comments to the Author

Reviewer #1: Thank you for the opportunity to review this interesting study on the relationship between financial hardship and patient-reported symptoms and clinical outcomes. In their study of patients with kidney failure on dialysis from two regional hospitals in Hong Kong, the authors find that unemployed patients and patients living below the poverty line experience worse clinical outcomes on composite measures and greater ER attendance relative to their employed peers and those living above the poverty line. They also note differences in certain symptoms; however, these differed by the measure of financial hardship.

This study is valuable and its contribution to the literature can be enhanced. I provide suggestions that I hope will improve the manuscript.

First, the setting of economic inequality in Hong Kong and the specific experience of healthcare-related financial hardship deserves greater attention. Hong Kong has enormous wealth disparity and housing segregation that may not be fully captured by employment and income. Further, the context of financial hardship as related to out-of-pocket healthcare costs requires further explanation given that public healthcare, as I understand, is free. It is important to understand both how this issue should be understood within the local context of Hong Kong and how it can be generalized to other global and health contexts. Second, the methodology and analysis are unclear. How were the instruments chosen? How was the subset of “patient-reported outcomes” reported in the table selected? Do they derive from the Dialysis Symptom Index, which includes a total of 30 items? This is not well explained. Also, how was the decision made to conduct so many t-tests? I am concerned that these multiple measures increase the likelihood of type I error. Third and finally, I believe the discussion overgeneralizes the interpretation of the results. The statements that “patient-reported and clinical outcomes differ between patients with different financial statuses in terms of their employment and income level” and financial “hardship may result in health inequity” do not appropriately account for the mixed results and study limitations.

I provide more detailed feedback below:

Title/Abstract

• Line 5/20: Recommend using “kidney failure” rather than “end-stage kidney disease” per guidelines advanced by Levey et al. (2020)

Levey AS, Eckardt KU, Dorman NM, Christiansen SL, Cheung M, Jadoul M, et al. Nomenclature for kidney function and disease: executive summary and glossary from a Kidney Disease: Improving Global Outcomes consensus conference*. Nephrology Dialysis Transplantation. 2020 Jul 1;35(7):1077–84.

Introduction

• Line 48: The authors state that patients may be required to cover 12-71% of dialysis costs out of pocket, but do not specify the nature of reimbursement at the hospitals in the study. This context would be especially helpful for understanding the impact of financial hardship on study participants. Also, I think it may be relevant to mention here that, as I understand it, public healthcare is virtually free and guaranteed in Hong Kong as this is not the case elsewhere.

• Lines 53-60: This section seems to emphasize the relationship between financial hardship and medical expenses, but I think the experience of economic oppression is more profound than that. I think the authors can do more to describe the local context of economic inequality in Hong Kong. I am not an expert in the region, though I think some relevant dynamics include issues of financialization, housing and spatial segregation, and immigration issues.

Materials and Methods

• Line 71: Are you able to add details about the hospitals from which these data were collected? They are described as “regional hospitals” but are they public? Private?

• Line 72: Please provide a citation where the inclusion and exclusion criteria are previously reported.

• Lines 93-100: I am not sure that the Methods adequately account for all four instruments and analyses. I also imagine chi-squared analyses were conducted to assess for differences in patient demographics, but these are not reported in the Methods. Looking at Table 1, I see composite scores for the Kidney Disease Quality of Life-36, the Karnofsky Performance Scale, and the Charlson Comorbidity Index, in addition to several individual patient-reported outcomes. Do the patient-reported outcomes derive from the Dialysis Symptoms Index? If so, how and why were only a subsample of the 30 symptoms selected for reporting?

Results

• The authors report significant results but do not describe results that are not significant, which is important.

Discussion

• Lines 126-128: I am not sure that this is the most accurate summary statement given that (1) your results differ based on employment status and poverty level, and (2) not all patient-reported outcomes were significantly different between your groups. I would revise this summary statement to provide a more cautious interpretation of your results.

• Lines 132-133: Consider rephrasing to situate the statistics of your study population relative to the general population (e.g., “In this study, roughly half as many people were employed and three times as many lived below the poverty line relative to the general population of Hong Kong”).

• Lines 137-142: Do you think that people experiencing economic oppression are at higher risk of developing kidney failure or that undergoing hemodialysis impedes employment opportunities, which then leads to financial hardship? Some literature on this might be helpful to include in the introduction. This might also be worth mentioning in the limitations with respect to the need for longitudinal analyses.

• Lines 159-160: Can you recommend additional factors affecting financial hardship (e.g., wealth, homeownership)?

Table 1

• The origin of the patient-reported outcomes is not clear to me. Are these from the DSI? If so, why are there not 30?

• What is the KDQOL-36 PCS vs. MCS?

• How did you choose serum albumin vs. urine microalbumin as a clinical outcome?

6. PLOS authors have the option to publish the peer review history of their article (what does this mean?). If published, this will include your full peer review and any attached files.

Reviewer #1: **Yes: **Jessica P Cerdeña, PhD

---

## [Author Response · Author response to Decision Letter 0]

5 Jan 2023

We are pleased to re-submit the revised manuscript entitled ‘Health Inequity Associated with Financial Hardship Among Patients with Kidney Failure: A Secondary Analysis’ (Manuscript No.: PONE-D-22-21227). We would like to thank the Editor and Reviewer for their thoughtful comments on the manuscript. Our responses to each of the comments are provided below in italics. The suggested revisions have strengthened the report of preliminary evidence, which informs directions for future research in health equity and financial wellbeing of patients with kidney failure. Thank you for your attention and I look forward to hearing from you.

Title/Abstract

• Line 5/20: Recommend using “kidney failure” rather than “end-stage kidney disease” per guidelines advanced by Levey et al. (2020)

Levey AS, Eckardt KU, Dorman NM, Christiansen SL, Cheung M, Jadoul M, et al. Nomenclature for kidney function and disease: executive summary and glossary from a Kidney Disease: Improving Global Outcomes consensus conference*. Nephrology Dialysis Transplantation. 2020 Jul 1;35(7):1077–84.

Response: We revised the wordings (e.g., kidney failure, kidney care) accordingly.

Introduction

• Line 48: The authors state that patients may be required to cover 12-71% of dialysis costs out of pocket, but do not specify the nature of reimbursement at the hospitals in the study. This context would be especially helpful for understanding the impact of financial hardship on study participants. Also, I think it may be relevant to mention here that, as I understand it, public healthcare is virtually free and guaranteed in Hong Kong as this is not the case elsewhere.

Response: We added a description about Hong Kong’s healthcare system in the Introduction section (p.4 lines 72-79).

• Lines 53-60: This section seems to emphasize the relationship between financial hardship and medical expenses, but I think the experience of economic oppression is more profound than that. I think the authors can do more to describe the local context of economic inequality in Hong Kong. I am not an expert in the region, though I think some relevant dynamics include issues of financialization, housing and spatial segregation, and immigration issues.

Response: Thanks for this very thoughtful comment. We added a description about Hong Kong’s economic inequity in the Introduction section (p.4 lines 72-79).

Materials and Methods

• Line 71: Are you able to add details about the hospitals from which these data were collected? They are described as “regional hospitals” but are they public? Private?

Response: We provided details about these hospitals in the Materials and Methods section (p.4 lines 82-84).

• Line 72: Please provide a citation where the inclusion and exclusion criteria are previously reported.

Response: Details of the original study can be found in the reference #16. We added the inclusion and exclusion criteria in the Methods section for better understanding of the study design (p.4 line 85-87).

• Lines 93-100: I am not sure that the Methods adequately account for all four instruments and analyses. I also imagine chi-squared analyses were conducted to assess for differences in patient demographics, but these are not reported in the Methods. Looking at Table 1, I see composite scores for the Kidney Disease Quality of Life-36, the Karnofsky Performance Scale, and the Charlson Comorbidity Index, in addition to several individual patient-reported outcomes. Do the patient-reported outcomes derive from the Dialysis Symptoms Index? If so, how and why were only a subsample of the 30 symptoms selected for reporting?

Response: We supplemented detailed accounts and references for the four instruments used in the Methods section (p.5 lines 92-110). To provide a fair comparison, we presented the results of all symptoms in the Dialysis Symptoms Index and other instruments in Table 1.

Results

• The authors report significant results but do not describe results that are not significant, which is important.

Response: Thanks for pointing out this issue. We stated non-significant results in the Results and Discussion sections (p.6 lines 138-139; p.7 lines 148-149) and included the statistics in the Table 1.

Discussion

• Lines 126-128: I am not sure that this is the most accurate summary statement given that (1) your results differ based on employment status and poverty level, and (2) not all patient-reported outcomes were significantly different between your groups. I would revise this summary statement to provide a more cautious interpretation of your results.

Response: We revised the summary to precisely capture both significant and non-significant results (p.7 lines 148-152).

• Lines 132-133: Consider rephrasing to situate the statistics of your study population relative to the general population (e.g., “In this study, roughly half as many people were employed and three times as many lived below the poverty line relative to the general population of Hong Kong”).

Response: We revised the statements based on your suggestion (p.7 line 154).

• Lines 137-142: Do you think that people experiencing economic oppression are at higher risk of developing kidney failure or that undergoing hemodialysis impedes employment opportunities, which then leads to financial hardship? Some literature on this might be helpful to include in the introduction. This might also be worth mentioning in the limitations with respect to the need for longitudinal analyses.

Response: Thanks for your suggestion. We added this point about disparities in the Introduction section (p.3 lines 53-55). We also stressed the need for a longitudinal study to investigate the impact of financial hardship on the outcomes of patients with kidney failure (p.8 lines 181-183).

• Lines 159-160: Can you recommend additional factors affecting financial hardship (e.g., wealth, homeownership)?

Response: We made recommendations on additional factors based on our recent review (p.8 lines 180-181).

Table 1

• The origin of the patient-reported outcomes is not clear to me. Are these from the DSI? If so, why are there not 30?

Response: We substantially revised Table 1 to present all the results, regardless their statistical significance levels.

• What is the KDQOL-36 PCS vs. MCS?

Response: We are sorry about the confusion. We defined all the abbreviations in the legends and provided details of the instruments in the Methods section (p.5 lines 92-110).

• How did you choose serum albumin vs. urine microalbumin as a clinical outcome?

Response: Serum albumin was chosen as an indicator of nutritional status but not disease progression. The data were retrieved from the laboratory results in the medical records. We analysed glomerular filtration rate to evaluate disease progression but no significant result was found (Table 1).

---

## [Decision Letter · Decision Letter 1]

27 Jan 2023

PONE-D-22-21227R1Health Inequity Associated with Financial Hardship Among Patients with Kidney Failure: A Secondary AnalysisPLOS ONE

Dear Dr. Ng,

Thank you for submitting your manuscript to PLOS ONE. After careful consideration, we feel that it has merit but does not fully meet PLOS ONE’s publication criteria as it currently stands. Therefore, we invite you to submit a revised version of the manuscript that addresses the points raised during the review process.

We look forward to receiving your revised manuscript.

Kind regards,

Ari Samaranayaka, PhD

Academic Editor

PLOS ONE

Additional Editor Comments (if provided):

Dear authors,

As the new academic editor, this is my first opportunity to read this manuscript, therefore it is likely to have points in my comments below that were not identified in the earlier review round.

1). Authors have used ttest to compare outcomes between two financial level groups. ttest does not account for confounders, therefore factors that authors identified should be interpreted only as markers not as those with independent effects. Therefore conclusions need to be cautious to be within that limitation.

2). Line 59/60. “Negative health outcomes, including depression, anxiety, lower health-related quality of life (HRQoL), and higher mortality risk, have been reported (9-11)”. Is this statement referring to those with lower financial levels or those with CKD in general?

3). Line 81. How were patients selected? All the patients who met inclusion/exclusion criteria in these 2 hospitals during the recruitment period were included or was there any selection? How representative the participants to non-participants if there was a selection? Was there any reason (like sample size estimation) for selecting specifically N=354 patients?

4). Line 87-88. “… research assistant administered a questionnaire…”. Please indicate what information were collected through this questionnaire. I assume this questionnaire was used to collect all patient-reported outcomes as opposed to assessing inclusion/exclusion criteria.

5). Line 82. “… patients were recruited from two public hospitals”. Could you name these hospitals?

6). Line 89/90. “… approved by the institutional research boards of the university and the involved hospitals”. Could you name these institutions?

7). Line 108-112. Please give references for weighted score version of Charlson Comorbidity Index and MDRD equation. Cited ref 23 not applicable for these measures.

8). Line 124. 58.5% were male. Different percentage in table1.

9). Line 131-141. This paragraph included results of comparing individual symptoms used to derive DSI score. This is a concern for multiple reasons. First, DSI index has been validated as a summative measure, not for individual symptoms, as reported in lines 92-110. Terefore how correct this comparison of individual items? Why validated summative DSI score not compared between groups? Second, ttest need outcome measures to be in continuous scale, patient-reported responses for these individual symptoms are in categorical scales (ie, likert scale coded to numeric) rather than in continuous scale, therefore not suitable for ttest. If authors need to retain these individual symptoms comparisons as results they need a justification with supporting reference(s) on why ttest is suitable. As at present ttest is not an appropriate statistical method. Otherwise comparison has to be done using a statistical method appropriate for the data. Same comment applicable to the corresponding results in table1.

10). Abstract line 32. “... increased distress associated with specific symptoms,…”. Methods section says dialysis symptom index (DSI) is derived from kidney symptoms, and higher DSI indicates higher distress. If so, above statement is obvious by definition, therefore I wonder why it is worth reporting as a result.

11). Table1. Please remove all asterisks and associated footnote because they are redundant.

12). Table1. Please check the correctness of the reported SD (0.11) for age for unemployed group.

13). Table1. Please mention Clinical visits and ER attendance are counted over what period. I could not find that in methods section. Please make sure method section includes how each of the reported measure was measured.

Reviewers' comments:

Reviewer's Responses to Questions

**Comments to the Author**

1. If the authors have adequately addressed your comments raised in a previous round of review and you feel that this manuscript is now acceptable for publication, you may indicate that here to bypass the “Comments to the Author” section, enter your conflict of interest statement in the “Confidential to Editor” section, and submit your "Accept" recommendation.

Reviewer #1: (No Response)

2. Is the manuscript technically sound, and do the data support the conclusions?

Reviewer #1: Yes

3. Has the statistical analysis been performed appropriately and rigorously? 

Reviewer #1: Yes

4. Have the authors made all data underlying the findings in their manuscript fully available?

Reviewer #1: Yes

5. Is the manuscript presented in an intelligible fashion and written in standard English?

Reviewer #1: Yes

6. Review Comments to the Author

Reviewer #1: Thank you for the opportunity to re-review this manuscript. I think this manuscript has improved significantly and I have just a few additional suggestions that I hope will strengthen it.

- It is not immediately clear to me why patients with active psychiatric disorders were excluded. Without going into too much detail, could the authors briefly explain this rationale? Also, the authors could also consider adding a flow diagram to demonstrate how they arrived at the final study population, unless this is included in previously published studies.

- I may misunderstand how the Dialysis Symptom Index should be used. I interpreted it as a summative instrument by which the positive response to more items indicates increased symptom burden. I am not sure what it means to interpret each symptom in isolation. For instance, what does it mean clinically that there is a significant difference in skin dryness and sleep maintenance between economically oppressed and advantaged people? I think it would be more meaningful to know whether there is a difference in symptom burden (or distress, as you write). This may be a null finding, which I think is understandable.

- I think your more meaningful finding is the significant difference in the Charleston Comorbidity Index. I think some of your discussion can be reframed to emphasize this specific finding—rather than the individual symptom differences—in addition to your other findings on healthcare utilization. For instance, I came across this older paper that looked into this question, and I imagine there are several more.

Droomers, Mariël, and Gert P. Westert. 2004. “Do Lower Socioeconomic Groups Use More Health Services, Because They Suffer from More Illnesses?” European Journal of Public Health 14 (3): 311–13. https://doi.org/10.1093/eurpub/14.3.311.

I recommend highlighting this finding as a key takeaway of the paper.

Again, thank you for the opportunity to review this paper and I look forward to seeing it published.

7. PLOS authors have the option to publish the peer review history of their article (what does this mean?). If published, this will include your full peer review and any attached files.

Reviewer #1: **Yes: **Jessica P. Cerdeña

---

## [Author Response · Author response to Decision Letter 1]

23 Feb 2023

We are pleased to re-submit the revised manuscript entitled ‘Health Inequity Associated with Financial Hardship Among Patients with Kidney Failure: A Secondary Analysis’ (Manuscript No.: PONE-D-22-21227). We would like to thank the Editor and Reviewer for their thoughtful comments on the manuscript. Our responses to each of the comments are provided below in italics. The suggested revisions have strengthened the report of preliminary evidence, which informs directions for future research in health equity and financial wellbeing of patients with kidney failure. Thank you for your attention and I look forward to hearing from you.

1). Authors have used ttest to compare outcomes between two financial level groups. ttest does not account for confounders, therefore factors that authors identified should be interpreted only as markers not as those with independent effects. Therefore conclusions need to be cautious to be within that limitation.

Response: Thanks for this kindly advice. The limitation related to the statistical methods was highlighted in the Discussion section.

2). Line 59/60. “Negative health outcomes, including depression, anxiety, lower health-related quality of life (HRQoL), and higher mortality risk, have been reported (9-11)”. Is this statement referring to those with lower financial levels or those with CKD in general?

Response: The statement was revised to enhance clarity.

3). Line 81. How were patients selected? All the patients who met inclusion/exclusion criteria in these 2 hospitals during the recruitment period were included or was there any selection? How representative the participants to non-participants if there was a selection? Was there any reason (like sample size estimation) for selecting specifically N=354 patients?

Response: Further details about recruitment were provided in the Methods and Results sections.

4). Line 87-88. “… research assistant administered a questionnaire…”. Please indicate what information were collected through this questionnaire. I assume this questionnaire was used to collect all patient-reported outcomes as opposed to assessing inclusion/exclusion criteria.

Response: The statement was revised to enhance clarity.

5). Line 82. “… patients were recruited from two public hospitals”. Could you name these hospitals?

Response: The names of the hospitals were provided.

6). Line 89/90. “… approved by the institutional research boards of the university and the involved hospitals”. Could you name these institutions?

Response: This statement which is redundant was removed. The names of the IRBs involved were provided together with the reference numbers in the following statement.

7). Line 108-112. Please give references for weighted score version of Charlson Comorbidity Index and MDRD equation. Cited ref 23 not applicable for these measures.

Response: References were updated for the captioned instruments.

8). Line 124. 58.5% were male. Different percentage in table1.

Response: Thanks for spotting out the inconsistency. The figure was checked against raw data and the numbers in Table 1 were revised.

9). Line 131-141. This paragraph included results of comparing individual symptoms used to derive DSI score. This is a concern for multiple reasons. First, DSI index has been validated as a summative measure, not for individual symptoms, as reported in lines 92-110. Therefore how correct this comparison of individual items? Why validated summative DSI score not compared between groups? Second, ttest need outcome measures to be in continuous scale, patient-reported responses for these individual symptoms are in categorical scales (ie, likert scale coded to numeric) rather than in continuous scale, therefore not suitable for ttest. If authors need to retain these individual symptoms comparisons as results they need a justification with supporting reference(s) on why ttest is suitable. As at present ttest is not an appropriate statistical method. Otherwise comparison has to be done using a statistical method appropriate for the data. Same comment applicable to the corresponding results in table1.

Response: Thanks for raising the statistical concerns. First, we added the comparison of DSI total scores in Table 1 for reference. The DSI not only generates a score to indicate the overall symptom burden, but it also provides a framework for assessing CKD-related symptoms, as demonstrated by good content validity and test-retest reliability of individual items in the original instrument (Weisbord et al., 2004). Of note, in a study by Weisbord et al. (2007), symptoms reported by patients and clinicians were compared based on the mean scores of individual symptoms. Second, it is implicitly assumed that the individual item scores of DSI are of interval scale data, otherwise it does not make sense to calculate a summative score for DSI. Indeed, it is common to report means and standard deviations of individual symptoms in previous studies (Almutary et al., 2016; de Rooji et al., 2022; Weisbord et al., 2007) and the item scores are not skewed. Therefore, we believe that t-test is appropriate to compare the severity levels of symptoms between groups.

References:

Almutary, H., Bonner, A., & Douglas, C. (2016). Which patients with chronic kidney disease have the greatest symptom burden? A comparative study of advanced CKD stage and dialysis modality. Journal of Renal Care, 42(2), 73-82.

de Rooij, E. N., Meuleman, Y., de Fijter, J. W., Jager, K. J., Chesnaye, N. C., Evans, M., ... & Hoogeveen, E. K. (2022). Symptom burden before and after dialysis initiation in older patients. Clinical Journal of the American Society of Nephrology, 17(12), 1719-1729.

Weisbord, S. D., Fried, L. F., Arnold, R. M., Rotondi, A. J., Fine, M. J., Levenson, D. J., & Switzer, G. E. (2004). Development of a symptom assessment instrument for chronic hemodialysis patients: The Dialysis Symptom Index. Journal of Pain and Symptom management, 27(3), 226-240.

Weisbord, S. D., Fried, L. F., Mor, M. K., Resnick, A. L., Unruh, M. L., Palevsky, P. M., ... & Arnold, R. M. (2007). Renal provider recognition of symptoms in patients on maintenance hemodialysis. Clinical Journal of the American Society of Nephrology, 2(5), 960-967.

10). Abstract line 32. “... increased distress associated with specific symptoms,…”. Methods section says dialysis symptom index (DSI) is derived from kidney symptoms, and higher DSI indicates higher distress. If so, above statement is obvious by definition, therefore I wonder why it is worth reporting as a result.

Response: The meaning of the captioned phrase means distress ‘originated from specific symptoms.’ We did not imply a statistical association between distress and specific symptoms. The statement was revised to enhance clarity.

11). Table1. Please remove all asterisks and associated footnote because they are redundant.

Response: All asterisks and footnotes were removed accordingly.

12). Table1. Please check the correctness of the reported SD (0.11) for age for unemployed group.

Response: The figures were checked against raw data and the number in the Table 1 was revised.

13). Table1. Please mention Clinical visits and ER attendance are counted over what period. I could not find that in methods section. Please make sure method section includes how each of the reported measure was measured.

Response: The duration of capturing clinical/ER visits and lengths of hospital stay were added in the Methods section and Table 1.

---

## [Decision Letter · Decision Letter 2]

14 Apr 2023

PONE-D-22-21227R2Health Inequity Associated with Financial Hardship Among Patients with Kidney Failure: A Secondary AnalysisPLOS ONE

Dear Dr. Ng,

Thank you for submitting your manuscript to PLOS ONE. After careful consideration, we feel that it has merit but does not fully meet PLOS ONE’s publication criteria as it currently stands. Therefore, we invite you to submit a revised version of the manuscript that addresses the points raised during the review process.

We look forward to receiving your revised manuscript.

Kind regards,

Mohammad Meshbahur Rahman, MS.

Academic Editor

PLOS ONE

Journal Requirements:

Reviewers' comments:

Reviewer's Responses to Questions

**Comments to the Author**

1. If the authors have adequately addressed your comments raised in a previous round of review and you feel that this manuscript is now acceptable for publication, you may indicate that here to bypass the “Comments to the Author” section, enter your conflict of interest statement in the “Confidential to Editor” section, and submit your "Accept" recommendation.

Reviewer #2: (No Response)

2. Is the manuscript technically sound, and do the data support the conclusions?

Reviewer #2: Yes

3. Has the statistical analysis been performed appropriately and rigorously? 

Reviewer #2: No

4. Have the authors made all data underlying the findings in their manuscript fully available?

Reviewer #2: No

5. Is the manuscript presented in an intelligible fashion and written in standard English?

Reviewer #2: Yes

6. Review Comments to the Author

Reviewer #2: The study titled “Health Inequity Associated with Financial Hardship Among Patients with Kidney Failure: A Secondary Analysis” aims to identify the differences in patient-reported and clinical outcomes among patients with different financial status. This is a very timely article with the distinct merit of linking patients outcomes and financial status. However, I have some observations as follows:

My other comments are as follows:

Abstract:

1. Line 22: Please specify the terms “kidney failure patients” in this line “This cross-sectional study aimed to identify the differences in patient-reported and clinical outcomes among patients with different financial status”

2. Line 24: Please start with the word in this line “354 patients with end-stage kidney disease were recruited from 25 March to June 2017 at two regional hospitals in Hong Kong”

Introduction:

3. Line 66: it may be "aid" instead of "aids."

4. Line 68: Please add this line “In Hong Kong, not enough research is done to examine health consequences of financial status.”

5. Line 73-75: Can the author give more recent data?

Materials and Methods:

6. Line 92: “After explaining the study and obtaining informed consent, the research assistant administered a questionnaire containing a demographic form and the instruments.” Please share that form as a supplementary file.

7. Line 108: it maybe "It’s" instead of "Its"

8. Line 110-118: Please specify categories of all outcome (categorical) variables and also how they are categorized.

9. Line 127-128: Please add a few lines about how you measure the P-value for frequency or percentage information. Did the author check the chi-square test for the categorical variables? Please provide the relevant statistics.

10. Line 166: Why the author didn’t conduct any regression model? Any explanation?

Results

11. Line 131: Please provide this information “58.5% were male” either in a graph or in the table.

Discussion:

12. Line 173-175: Can the author justify this?

13. Line 179-182: Please clarify “While symptoms and other outcomes are signs of deteriorating health, patients with a poorer symptom status or poorer outcomes may have a higher demand for healthcare services and a lower physical capacity for engaging actively in employment.”

7. PLOS authors have the option to publish the peer review history of their article (what does this mean?). If published, this will include your full peer review and any attached files.

Reviewer #2: **Yes: **Mohammad Nayeem Hasan

---

## [Author Response · Author response to Decision Letter 2]

18 Apr 2023

We are pleased to re-submit the revised manuscript entitled ‘Health Inequity Associated with Financial Hardship Among Patients with Kidney Failure: A Secondary Analysis’ (Manuscript No.: PONE-D-22-21227). We would like to thank the Editor and Reviewer for their thoughtful comments on the manuscript. Our responses to each of the comments are provided below in italics. The suggested revisions have strengthened the report of preliminary evidence, which informs directions for future research in health equity and financial wellbeing of patients with kidney failure. Thank you for your attention and I look forward to hearing from you.

1. Line 22: Please specify the terms “kidney failure patients” in this line “This cross-sectional study aimed to identify the differences in patient-reported and clinical outcomes among patients with different financial status”

Response: The sentence was revised accordingly.

2. Line 24: Please start with the word in this line “354 patients with end-stage kidney disease were recruited from 25 March to June 2017 at two regional hospitals in Hong Kong”

Response: The sentence was revised so that it begins with a word.

3. Line 66: it may be "aid" instead of "aids."

Response: The term was revised accordingly.

4. Line 68: Please add this line “In Hong Kong, not enough research is done to examine health consequences of financial status.”

Response: We inserted a statement based on this suggestion.

5. Line 73-75: Can the author give more recent data?

Response: We updated the latest available figures of per capita GDP and Gini coefficient in 2021 and 2019 (World Bank Group, 2023; World Economics, 2019), respectively.

References

World Bank Group. GDP per capita (current US$) – Hong Kong SAR, China [Internet]. 2023 [Accessed 18 April 2023]. Available: https://data.worldbank.org/indicator/NY.GDP.PCAP.CD?locations=HK.

World Economics. Hong Kong’s Gini coefficient [Internet]. 2019 [Accessed 18 April 2023]. Available from: https://www.worldeconomics.com/Inequality/Gini-Coefficient/Hong%20Kong.aspx#.

6. Line 92: “After explaining the study and obtaining informed consent, the research assistant administered a questionnaire containing a demographic form and the instruments.” Please share that form as a supplementary file.

Response: We attached the demographic form as a supplementary material in the revised version.

7. Line 108: it maybe "It’s" instead of "Its"

Response: We checked with the editor and confirmed that the use of ‘its’ in this case is appropriate.

8. Line 110-118: Please specify categories of all outcome (categorical) variables and also how they are categorized.

Response: The outcomes (i.e., functional status, comorbidity level, healthcare service utilization, biochemical parameters) were analysed as continuous variables and their measurements were described in the paragraph. We have now specified all categories of the categorical variables (i.e., sex, marital status, education level, dialysis modality, history of transplantation) in Table 1.

9. Line 127-128: Please add a few lines about how you measure the P-value for frequency or percentage information. Did the author check the chi-square test for the categorical variables? Please provide the relevant statistics.

Response: We added detailed description about comparing background characteristics of patients with different financial status. Specifically, categorical variables were compared between the financial status groups by using chi-squared test, while all patient-reported and clinical outcomes were compared by using independent sample t-test.

10. Line 166: Why the author didn’t conduct any regression model? Any explanation?

Response: This study analysed data from the original study on symptom experience of patients with kidney failure. Factors potentially influencing financial hardship (e.g., financial aids, home ownership, personal savings, health spendings) have not been accessed in the original study (Ng et al., 2020; 2021). It is therefore immature to conduct adjusted analysis, such as regression analysis. Exploratory univariate analyses were only conducted aiming to provide some insights into whether there are differences in patient-reported and clinical outcomes in terms of financial hardship. 

References

Ng MSN, Miaskowski C, Cooper B, Hui YH, Ho EHS, Mo SKL, et al. Distinct symptom experience among subgroups of patients with ESRD receiving maintenance dialysis. J Pain Symptom Manage 2020;60:70-9.

Ng MSN, Chan DNS, Cheng Q, Miaskowski C, So WKW. Association between financial hardship and symptom burden in patients receiving maintenance dialysis: a systematic review. Int J Environ Res Public Health 2021;18:9541.

11. Line 131: Please provide this information “58.5% were male” either in a graph or in the table.

Response: This information was presented in Table 1 (Background characteristics – Overall).

12. Line 173-175: Can the author justify this?

Response: We added a sentence to clarify the relationship with the support of a relevant study.

13. Line 179-182: Please clarify “While symptoms and other outcomes are signs of deteriorating health, patients with a poorer symptom status or poorer outcomes may have a higher demand for healthcare services and a lower physical capacity for engaging actively in employment.”

Response: The sentence was revised to clarify the reciprocal relationship. While symptoms and other outcomes may be the consequences of financial hardship, they may also reflect a lower physical capacity for engaging in economic activities.

---

## [Decision Letter · Decision Letter 3]

29 May 2023

PONE-D-22-21227R3Health Inequity Associated with Financial Hardship Among Patients with Kidney Failure: A Secondary AnalysisPLOS ONE

Dear Dr. Ng,

Thank you for submitting your manuscript to PLOS ONE. After careful consideration, we feel that it has merit but does not fully meet PLOS ONE’s publication criteria as it currently stands. Therefore, we invite you to submit a revised version of the manuscript that addresses the points raised during the review process.

We look forward to receiving your revised manuscript.

Kind regards,

Mohammad Meshbahur Rahman, MS.

Academic Editor

PLOS ONE

Reviewers' comments:

Reviewer's Responses to Questions

**Comments to the Author**

1. If the authors have adequately addressed your comments raised in a previous round of review and you feel that this manuscript is now acceptable for publication, you may indicate that here to bypass the “Comments to the Author” section, enter your conflict of interest statement in the “Confidential to Editor” section, and submit your "Accept" recommendation.

Reviewer #3: (No Response)

2. Is the manuscript technically sound, and do the data support the conclusions?

Reviewer #3: Partly

3. Has the statistical analysis been performed appropriately and rigorously? 

Reviewer #3: No

4. Have the authors made all data underlying the findings in their manuscript fully available?

Reviewer #3: Yes

5. Is the manuscript presented in an intelligible fashion and written in standard English?

Reviewer #3: Yes

6. Review Comments to the Author

Reviewer #3: The authors studied an important issue in relation to chronic kidney disease (CKD) or kidney failure. Financial hardship and inability to pay often leads to kidney failure receiving inadequate or suboptimal care. However, the authors used two variables to indirectly read the financial hardship, namely, employment and monthly family income.

Based on the suggestions by the previous reviewers, authors have modified the manuscript and have properly answered the points except the question regarding why regression analysis was not carried out.

Although the authors tried to give an explanation, which is not quite satisfactory. I think a logistic or linear regression analysis was possible by dichotomizing the KPS scale scores or taking it as a continuous scale, where important factors age, sex, marital status, employment, education years, dialysis types, history of transplantation, days of hospital stay could have been taken as factor variables, in order to check the nature and magnitude of association of financial hardship and health when adjusted for other factors. Please note that if you find some patients unemployed but having good financial status, this could indirectly indicate that they are either receiving aid / or using personal savings. Similarly the health care service utilization and its relationship with various factors could checked in a similar way.

In addition, from the methods it seems clear that some information was collected directly from patients. Hence, the words “A secondary analysis” in the title does not seem appropriate.

The author should explore the said analyses, add additional tables and description and add necessary discussion.

7. PLOS authors have the option to publish the peer review history of their article (what does this mean?). If published, this will include your full peer review and any attached files.

Reviewer #3: No

---

## [Author Response · Author response to Decision Letter 3]

31 May 2023

We are pleased to re-submit the revised manuscript entitled ‘Health Inequity Associated with Financial Hardship Among Patients with Kidney Failure’ (Manuscript No.: PONE-D-22-21227). We would like to thank the Editor and Reviewer for their thoughtful comments on the manuscript. Our responses to each of the comments are provided below in italics. The suggested revisions have strengthened the report of preliminary evidence, which informs directions for future research in health equity and financial wellbeing of patients with kidney failure. Thank you for your attention and I look forward to hearing from you.

1. Although the authors tried to give an explanation, which is not quite satisfactory. I think a logistic or linear regression analysis was possible by dichotomizing the KPS scale scores or taking it as a continuous scale, where important factors age, sex, marital status, employment, education years, dialysis types, history of transplantation, days of hospital stay could have been taken as factor variables, in order to check the nature and magnitude of association of financial hardship and health when adjusted for other factors. 

Response: We used multiple regression analyses to examine the associations between outcome variables, employment status, and income level adjusted for demographic factors (sex, marital status, education level, dialysis modality, history of transplantation, age, duration on dialysis). Findings were presented in Tables 1 and 2. The relevant contents in the Results, Discussion, and Abstract were amended accordingly.

2. In addition, from the methods it seems clear that some information was collected directly from patients. Hence, the words “A secondary analysis” in the title does not seem appropriate.

Response: We deleted the term ‘secondary analysis’ from the title and Methods.

3. The author should explore the said analyses, add additional tables and description and add necessary discussion.

Response: We conducted regression analyses, and their details and implications were added in the Methods, Results, and Discussions as appropriate.

---

## [Decision Letter · Decision Letter 4]

6 Jun 2023

PONE-D-22-21227R4Health Inequity Associated with Financial Hardship Among Patients with Kidney FailurePLOS ONE

Dear Dr. Ng,

Thank you for submitting your manuscript to PLOS ONE. After careful consideration, we feel that it has merit but does not fully meet PLOS ONE’s publication criteria as it currently stands. Therefore, we invite you to submit a revised version of the manuscript that addresses the points raised during the review process.

We look forward to receiving your revised manuscript.

Kind regards,

Mohammad Meshbahur Rahman, MS.

Academic Editor

PLOS ONE

Journal Requirements:

Reviewers' comments:

Reviewer's Responses to Questions

**Comments to the Author**

1. If the authors have adequately addressed your comments raised in a previous round of review and you feel that this manuscript is now acceptable for publication, you may indicate that here to bypass the “Comments to the Author” section, enter your conflict of interest statement in the “Confidential to Editor” section, and submit your "Accept" recommendation.

Reviewer #3: All comments have been addressed

2. Is the manuscript technically sound, and do the data support the conclusions?

Reviewer #3: Yes

3. Has the statistical analysis been performed appropriately and rigorously? 

Reviewer #3: Yes

4. Have the authors made all data underlying the findings in their manuscript fully available?

Reviewer #3: Yes

5. Is the manuscript presented in an intelligible fashion and written in standard English?

Reviewer #3: Yes

6. Review Comments to the Author

Reviewer #3: Thank you addressing the comments. The manuscript now looks complete. The recommended analyses has given some additional insights to the main findings.

Several minor corrections are recommended-

1. The following variables are not normally distributed in your data: Months on dialysis, CCI, clinical visits in 6 months, ER attendance in 6 months and days of hospital stay in 6 months. It is not justified to use t test for comparison of these variables across income and employment groups. Also, data should be expressed as 'median (IQR)' rather than as mean SD. So, for these variables in the tables do following things.

a. Use Wicoxon Rank Sum Test (Mann-Whiteny U test) for comparison

b. Express as Median (IQR) for all three columns

2. Instead of writing N % and Mean SD in the table rows, explain how data was presented in the footnotes. For example you can write- Data was presented as N %, Mean SD and Median IQR where appropriate.

3. Remove the column containing t values from the table. The p-values expressed in three decimal points are enough to display the student distribution of the differences.

4. Give the full forms of abbreviations used in the tables in table footnotes. (E.g. PCS, MCS, KPS, CCI etc).

7. PLOS authors have the option to publish the peer review history of their article (what does this mean?). If published, this will include your full peer review and any attached files.

Reviewer #3: **Yes: **Md. Abdullah Saeed Khan

---

## [Author Response · Author response to Decision Letter 4]

6 Jun 2023

We are pleased to re-submit the revised manuscript entitled ‘Health Inequity Associated with Financial Hardship Among Patients with Kidney Failure’ (Manuscript No.: PONE-D-22-21227). We would like to thank the Editor and Reviewer for their thoughtful comments on the manuscript. Our responses to each of the comments are provided below in italics. The suggested revisions have strengthened the report of preliminary evidence, which informs directions for future research in health equity and financial wellbeing of patients with kidney failure. Thank you for your attention and I look forward to hearing from you.

1. The following variables are not normally distributed in your data: Months on dialysis, CCI, clinical visits in 6 months, ER attendance in 6 months and days of hospital stay in 6 months. It is not justified to use t test for comparison of these variables across income and employment groups. Also, data should be expressed as 'median (IQR)' rather than as mean SD. So, for these variables in the tables do following things.

a. Use Wicoxon Rank Sum Test (Mann-Whiteny U test) for comparison

b. Express as Median (IQR) for all three columns

Response: We appreciate these precise and constructive suggestions. We conducted the analyses as advised and revised the relevant parts in the manuscript (including Analyses, Results, and Tables).

2. Instead of writing N % and Mean SD in the table rows, explain how data was presented in the footnotes. For example you can write- Data was presented as N %, Mean SD and Median IQR where appropriate.

Response: We added a footnote to indicate variables of which medians and inter-quartile ranges are presented.

3. Remove the column containing t values from the table. The p-values expressed in three decimal points are enough to display the student distribution of the differences.

Response: The column was deleted as advised.

4. Give the full forms of abbreviations used in the tables in table footnotes. (E.g. PCS, MCS, KPS, CCI etc).

Response: We provided the full names of the instruments in the Tables.

---

## [Decision Letter · Decision Letter 5]

7 Jun 2023

Health Inequity Associated with Financial Hardship Among Patients with Kidney Failure

PONE-D-22-21227R5

Dear Dr. Ng,

We’re pleased to inform you that your manuscript has been judged scientifically suitable for publication and will be formally accepted for publication once it meets all outstanding technical requirements.

Kind regards,

Mohammad Meshbahur Rahman, MS.

Academic Editor

PLOS ONE

Additional Editor Comments (optional):

Reviewers' comments:

Reviewer's Responses to Questions

**Comments to the Author**

1. If the authors have adequately addressed your comments raised in a previous round of review and you feel that this manuscript is now acceptable for publication, you may indicate that here to bypass the “Comments to the Author” section, enter your conflict of interest statement in the “Confidential to Editor” section, and submit your "Accept" recommendation.

Reviewer #3: All comments have been addressed

2. Is the manuscript technically sound, and do the data support the conclusions?

Reviewer #3: Yes

3. Has the statistical analysis been performed appropriately and rigorously? 

Reviewer #3: Yes

4. Have the authors made all data underlying the findings in their manuscript fully available?

Reviewer #3: Yes

5. Is the manuscript presented in an intelligible fashion and written in standard English?

Reviewer #3: Yes

6. Review Comments to the Author

Reviewer #3: This a good study discovering impacts of socioeconomic inequity in the care of complicated chronic disease. Best wishes for you hard work.

7. PLOS authors have the option to publish the peer review history of their article (what does this mean?). If published, this will include your full peer review and any attached files.

Reviewer #3: **Yes: **Md. Abdullah Saeed Khan

---

## [Editor Report · Acceptance letter]

13 Jun 2023

PONE-D-22-21227R5 

Health Inequity Associated with Financial Hardship Among Patients with
Kidney Failure 

Dear Dr. Ng:

I'm pleased to inform you that your manuscript has been deemed suitable for publication in PLOS ONE. Congratulations! Your manuscript is now with our production department. 

Kind regards, 

on behalf of

Mr. Mohammad Meshbahur Rahman 

Academic Editor

PLOS ONE